# Modulation of Placental Gene Expression in Small-for-Gestational-Age Infants

**DOI:** 10.3390/genes11010080

**Published:** 2020-01-10

**Authors:** Jessica L. O’Callaghan, Vicki L. Clifton, Peter Prentis, Adam Ewing, Yvette D. Miller, Elise S. Pelzer

**Affiliations:** 1School of Biomedical Sciences, Faculty of Health, Queensland University of Technology, Brisbane 4001, Queensland, Australia; e.pelzer@qut.edu.au; 2Institute of Health and Biomedical Innovation, Faculty of Health, Queensland University of Technology, Brisbane 4059, Queensland, Australia; 3Mater Medical Research Institute, University of Queensland, Brisbane 4101, Queensland, Australia; vicki.clifton@mater.uq.edu.au (V.L.C.); adam.ewing@mater.uq.edu.au (A.E.); 4School of Earth, Environmental and Biological Sciences, Science and Engineering Faculty, Queensland University of Technology, Brisbane 4001, Queensland, Australia; p.prentis@qut.edu.au; 5School of Public Health and Social Work, Faculty of Health, Queensland University of Technology, Brisbane 4059, Queensland, Australia; yvette.miller@qut.edu.au

**Keywords:** placenta, SGA, bacterial signatures, gene regulation

## Abstract

Small-for-gestational-age (SGA) infants are fetuses that have not reached their genetically programmed growth potential. Low birth weight predisposes these infants to an increased risk of developing cardiovascular, metabolic and neurodevelopmental conditions in later life. However, our understanding of how this pathology occurs is currently incomplete. Previous research has focused on understanding the transcriptome, epigenome and bacterial signatures separately. However, we hypothesise that interactions between moderators of gene expression are critical to understanding fetal growth restriction. Through a review of the current literature, we identify that there is evidence of modulated expression/methylation of the placental genome and the presence of bacterial DNA in the placental tissue of SGA infants. We also identify that despite limited evidence of the interactions between the above results, there are promising suggestions of a relationship between bacterial signatures and placental function. This review aims to summarise the current literature concerning fetal growth from multiple avenues and propose a novel relationship between the placental transcriptome, methylome and bacterial signature that, if characterised, may be able to improve our current understanding of the placental response to stress and the aetiology of growth restriction.

## 1. Introduction

The placenta is the hypomethylated, transitional organ that supports the growth of the developing fetus throughout gestation. The tightly regulated and timed addition of methylation marks and transcriptional activation is critical to normal fetal development [1]. This review investigates the interactions of moderators of placental gene expression in the context of fetal growth restriction and the potential role of bacterial DNA in the regulation process. Assessment of transitional, acute or chronic exposure of the fetus to bacteria and bacterial components during gestation has largely failed to demonstrate causality in adverse fetal outcomes. Recent technological advances have enabled a more integrative analysis of complex data [2,3]. We propose that a more integrative approach to data analysis is more representative of the complexity of a biological system, and that such investigations may lead to an improved understanding of the role of bacterial components in pregnancy outcomes.

## 2. Growth Restriction *in Utero*—Causes and Potential Aetiology

The pathological decrease in growth *in utero* can contribute to the onset of many lifelong health conditions. Children born with a birth weight in the 10th percentile for growth in Australia are considered growth restricted or small for gestational age (SGA) [4]. These births account for 30% of term infants born in low- and middle-income countries and up to 8% in developed countries [5]. These children may be constitutionally small and experience no further risk to their health, the same as their appropriate-for-gestational-age (AGA) counterparts or they may have an increased risk of developing metabolic (i.e. Type 2 diabetes [6]), cardiovascular (i.e., heart attacks and hypertension [7]), immune (i.e., the development of autoimmune diseases and asthma [8,9,10]), and even psychological (i.e., depressive disorders [11]) conditions later in life. This correlation is due to the developmental origins of disease hypothesis (DoHaD) [12]. This hypothesis states that unfavourable alterations to the *in utero* environment during gestation leads to fetal acclimation in order to retain proper growth and function in the short-term timeline. However, this acclimation can lead to complications later in life due to the initial differential function of the fetal organs. The most common fetal outcome that is linked to the DoHaD hypothesis is reduced birth weight. The understanding of the onset of the growth restriction is lacking. However, multiple risk factors have been identified. Previous experimental studies have characterised the contribution of maternal (habits such as smoking [13] and drinking [14,15] and maternal comorbidities such as maternal asthma [16]), fetal (chromosomal abnormalities and genetic metabolism conditions [17]) and placental (poor placental nutrient transfer [18] and poor angiogenesis [19]) factors which may have resulted in fetal growth restriction (FGR) [20,21]. Despite recent advances, the exact mechanisms in which the placenta, the mediator between mother and fetus, functions and attempts to acclimate in growth restriction remains unknown. As poor placental function is highly correlated with fetal growth restriction, it is critical to characterise the specific genetic pathways involved in SGA within the placenta itself and to further elucidate the internal and external factors that cause these pathways to become dysregulated.

## 3. Placental Contribution to Fetal Growth *in Utero* via Gene Expression

Placental gene expression changes are normal physiological responses to fetal growth and development. As the pregnancy progresses, the expression profiles of specific placental transcripts will alter to accommodate changing fetal needs. For example, expression of the gene encoding human chorionic gonadotrophin (hCG) is based on the gestational week of pregnancy and expression will rise and fall according to need [22]. The levels of hCG rise during the first trimester to maintain the secretion of progesterone from the ovary, a hormone that sustains enrichment of the uterine lining to allow placentation to occur [23]. This protein promotes angiogenesis and invasion of the placenta into the uterine lining, to allow proper growth of the fetus to occur. After ten to eleven weeks of gestation, hCG protein levels drop and are maintained at a lower protein abundance and gene expression level for the remainder of the pregnancy. The hCG protein is then redirected to perform other important roles such as growth and differentiation of fetal organs and expansion of the uterus to accommodate the growing fetus [24]. Pathological decreases or increases in gene expression of hCG and protein abundance especially in early pregnancy, however, can result in pregnancy complications including miscarriage [25,26]. Proper gene expression changes are therefore inherently essential for the pregnancy to progress and function and dysregulation of these processes can lead to pathological outcomes for the infant and/or mother.

Previously, sex-specific placental gene expression has been characterised in non-pathological pregnancies [27]. A study completed by Sood et al., 2006 [28] highlighted that there is a higher expression of immune related genes (i.e., *K1*, *IL2RB*, Clusterin, *LTBP* and *CXCL1*) in the placental tissue from pregnancies that yielded a female infant compared to a male infant. This sex-specific placental gene expression is conserved in other species as exhibited by the differential expression of transcripts based on fetal sex in response to changes to maternal diet in mice [29], rats [30], rabbits [31], baboons [32] and humans [33,34] indicting that there is a differential acclimation to *in utero* stress dependent on infant sex via differential placental function.

### Pathological Placental Gene Expression Changes Resulting in Growth-Restricted Infants

Dysregulated placental gene expression can result in fetal growth restriction in both male and female infants. An RNA-sequencing study was completed by Sõber et al., 2015 [35] using human placentae collected from pregnancies resulting in SGA infants with and without pre-eclampsia to characterise gene expression changes associated with fetal growth. This study identified the similar expression pattern of genes between late-onset pre-eclampsia (LO-PE) with intrauterine growth restriction (IUGR) (another term for pathologically growth-restricted infants) and SGA alone as well as LO-PE without IUGR to placentas from pregnancies where the infant was large for gestational age (LGA). It also characterised differential expression of transcripts related to growth, immune function, cell structure and blood clotting with and without the onset of another pathological condition (pre-eclampsia). An independent study completed by Deyssenroth et al. 2017 [36] sequenced the transcriptomes of 200 human placentas from pregnancies that resulted in SGA, LGA and AGA infants. This study reported that transcripts from functional groups including cell proliferation (*CREB3*), protein transport (*DNAJC14*), cellular growth (*DDX3X*), development (*GRHL1*) and proliferation (*C21orf91*) were increased in SGA placentae. A further RNA-sequencing study completed by Majewska et al., 2019 [37] undertook a transcriptomic analysis of IUGR placentas. These authors hypothesise that expression changes are contributing to the proinflammatory changes occurring in fetal growth restriction. Specific genes identified included *ARMS2*, *ADAM2* and *THEMIS*, which are involved in T cell regulation and immune response as well as inflammation. Recurring functional groups of enriched transcripts in both RNA-sequencing and microarray studies include cell proliferation (i.e., *CREB3*), cell adhesion (i.e., *PVRL4*), angiogenesis (i.e., *FLT1*, *ENG*), immune system (i.e., *PSG1*) and inflammation (i.e., *HPGD*), metabolism (i.e., *LEP*) and protein transport (i.e., *DNAJC14*) (See Table 1 for a summary of recent RNA-sequencing advances and Table 2 for microarray analysis).

Different environmental factors also contribute to a known sex-specific differential gene expression in pregnancy pathologies. In the proinflammatory state of maternal obesity, there is an upregulation of hypoxia-inducible factor 1-α-regulated miRNA 210 and tumor necrosis factor-alpha (TNF-α) mRNA in the placenta from female infants that correlated with decreased placental function [33]. In the same study there was a measured increase in male birth weight, potentially leading to male macrosomia (fetal overgrowth). Certain co-morbidities can also influence fetal birth weight in a sex-dependent manner including the presence of maternal asthma (outcomes discussed later in the article) and mild preeclampsia [38], where there is a correlation with growth reduction of females. Placental growth transcripts that are differentially expressed haven’t been stratified by stressor or infant sex indicating that there may be a bigger difference than currently identified and that multiple cofactors are involved in the regulation of fetal growth.

The lack of a single core dysregulated pathway that is shared across studies that is involved in growth restriction also suggests that global expression changes are associated with this condition. Current studies have utilised RNA-sequencing as a method for charactering the global gene expression changes in fetal growth restriction and these studies have identified multiple potential causal pathways. However, they have only just begun to understand the molecular changes associated with this condition and the specific gene expression changes that contribute to the onset of growth restriction. Further research is required using human tissue to characterise the placental transcriptome of SGA infants, correcting for multiple confounders including type of *in utero* stress, fetal sex and maternal characteristics.

The onset of pathological growth restriction may also be due to continued poor placental acclimation to constant insult compared to either the hypothesis that FGR is due to a genetic insufficiency or that one instance of in utero stress results in size reduction. The emerging pattern of multiple distinct pathway gene expression changes supports a role for the double hit hypothesis, whereby the placenta can acclimate to one insult by differentially expressing proteins but is impaired in response to a second. This is further explained by the placental ability to adapt to one pregnancy co-morbidity and the lack of acclimation to a further or continued insult. Another environmental variable that demonstrates this ‘double hit’ is in asthmatic mothers (first insult) whose exacerbations (the second insult) result in SGA fetuses [20]. The change in fetal weight is also sex dependent in asthma; during the first insult, female infants will reduce growth output to a smaller but not pathological level while the males continue down a normal growth trajectory. After the second hit, males are then challenged and their lack of compensation during the earlier stages of pregnancy results in their pathological growth restriction while females remain smaller but not SGA [20]. This is also confirmed by the hypothesis that males exhibit higher *in utero* vulnerability and lack of ability to acclimate to sub-optimal conditions [39], while, in females, the placenta has a protective function in adverse circumstances. The double hit hypothesis may be consistent with literature that places late-onset and early-onset pre-eclampsia (PE) in similar clusters to growth restriction in terms of expression of key transcripts compared to the healthy placenta. These placentas have potentially acclimated to PE which is evident in the shared expression profile with this condition but received further insult to the pregnancy resulting in SGA status due to poor compensation. This theory, however, requires further clarification and research to identify this shared acclimation in this subset of PE-IUGR infants. Additionally, a study exhibited that previous exposure of a pregnant mouse to a bacterial strain enhances the effects of later administered bacterial products (i.e., lipopolysaccharides from the Gram-negative bacterial cell wall) correlated with preterm birth [40]. The study proposed this mechanism as placental ability to protect against the first hit but not the second, leading to poor fetal outcome. The double hit hypothesis suggests that external exposures and modulators including lifestyle choices as well as genetics may be responsible for poor placental acclimation to insult. The poor placental function may then result in fetal growth restriction. This also proposes the potential for multiple molecular pathways to become dysregulated dependent on the specific placental acclimation to stress. The genetic pathways involved in placental changes affecting fetal growth are emerging; however, the intrinsic and extrinsic factors responsible for pathway dysregulation and the underlying mechanisms in the different pathways of fetal growth restriction remain largely unknown.

It is well accepted that gene expression changes induced by intrinsic and extrinsic modulators are regulated by epigenetic modifications of DNA or histone proteins. However, details of specific modifications involved in pathological fetal growth changes, placental gene expression and differential methylation resulting in, or due to SGA are poorly understood. However, placental epigenome changes may be contributing to expression changes via the remodeling of epigenetic marks on the genome in response to external stimuli. Methylation of the genome is such a mark and is frequently characterised in placenta pathologies.

## 4. Placental DNA Methylation Is Associated with Normal Fetal Growth

DNA methylation, the addition of a methyl group to a cytosine paired to a guanine (CpG) dinucleotide, in clusters located near or within specific genes at distinct time points, is required during gestation [47]. The specific methylation of loci in the placental genome is a reversible modification of DNA that results in the selective expression of specific genes and is associated with normal fetal growth [48]. Differential expression of genes is required as the placenta needs to acclimate during gestation in response to external environmental cues and developmental needs of the fetus. This allows the placenta to function adequately and to accommodate the growth of the fetus. For example, in a mouse model, knockdown of DNA (cytosine-5-)-methyltransferase 3-like (Dnmt3l), an enzyme with a critical role in adding and removing methyl groups, resulted in morphological defects of the placenta demonstrating the central role of methylation in normal tissue function [49].

An important function of methylation of the placental genome is imprinting. Genomic imprinting is the selective expression of only one of the two inherited alleles; either the maternal copy of the gene or the paternal copy of the gene is expressed. Currently, approximately 50 human genes have being identified as imprinted genes with the large majority of these characterised in placental tissue and are related to birth outcomes including fetal weight [50,51]. This may be due to the theory that there is a differential allocation of resources dependent on the expression of either maternally or paternally imprinted genes. Expression of maternally imprinted genes limits nutrients to the fetus, while the expression of paternally imprinted genes contributes to the increase in resources to the infant during gestation. It is the balance between giving and taking resources that allow the fetus to grow appropriately. Without normal placental function, there is a higher risk of poor fetal outcomes including SGA infants as well as preterm birth and stillbirth. Differential methylation of the placental epigenome is, therefore, a critical step in the development and growth of the fetus and, changes to this process may result in adverse outcomes.

### Placental DNA Methylation Associated with Altered Fetal Growth

Differential DNA methylation of specific loci in the placental genome has also been implicated in multiple pathologies including but not limited to fetal growth restriction [52]. Previous literature has characterised distinct methylation signatures that can be used to cluster acute-chorioamnionitis (methylation signature consistent with activation of immune cells and innate immune response) [53], early and late-onset pre-eclampsia [54], as well as preterm birth (gene ontology terms related to organ development and metabolic function) [55], as distinct molecular subtypes. Differential methylation of particular loci in pathological placenta has been identified. However, understanding of the exact effect of these methylation changes on the onset of fetal growth restriction is unknown.

Studies comparing the placental epigenome of women who have developed gestational diabetes mellitus have characterised differentially methylated genes involved in metabolic pathways including MAPK signaling and, cell adhesion and signaling. Studies have also correlated fetal birth weight and placental expression with changes in DNA methylation at the specific loci [56,57,58]. The involvement of multiple methylation changes in placental pathologies further demonstrates the crucial role that DNA methylation has in placental development and growth and the lack of a core dysregulated pathway in the onset of placenta associated fetal growth restriction.

Comparison of the placental methylation profile of growth-restricted infants to an appropriate-weight infant (Table 3) has identified multiple genes involved in a range of pathways. Commonly, *H19* and *IGF2*, examples of imprinted genes, are hypomethylated (loss of methylation) in the placenta of growth-restricted infants. The *H19* gene, maternally expressed (paternally imprinted), deprives the fetus of nutrients while the *IGF2*, paternally expressed (maternally imprinted) gene, increased the resource allocation to the infant [59]. However, these growth-related genes are not the sole pathway involved in this pathology. Multiple genes including inflammatory (i.e., *IL10*, *CD28*, *CD38*), cardiovascular (i.e., *ACE*, *NO53*, *CASZI*) and metabolic genes (i.e., *BLK*, *PTRN2*, *GK*) (Table 3) also have altered methylation status in the placental genome of SGA infants, suggesting that the expression differences stated in the previous section are mirrored in the DNA methylation of the genome. However, there is a lack of direct overlap even when comparing methylation and expression data from the same samples, which may be due to multiple epigenetic mechanisms (i.e., histone modifications and small interfering RNA (siRNA) interactions contributing to fetal growth changes). The role of methylation changes in fetal growth restriction should not be ignored, however, as the role of CpGs have a major contribution to fetal growth. Epigenetic changes are usually implicated in response to a stimulus which requires different cellular pathways than those currently expressed. However, the stimulus that induces the above remodeling in placental tissue is unknown.

## 5. Potential Modulators of DNA Methylation Associated with Altered Fetal Growth

DNA methylation changes in the placenta are usually the result of an adjustment to external modulators or stimuli in which the placenta must acclimate to produce a healthy infant. Characterised maternal clinical data that are correlated with changes in the placental epigenome and resulting offspring include maternal asthma [60], smoking [61], alcohol consumption [62], and maternal dietary supplements [63,64]. When methylation of *AluYb8,* a repetitive sequence found in the genome, is increased in human placenta, the fetal birth weight percentile also increases [65]. These methylation levels also differed significantly between infants who were exposed to cigarette smoke in utero compared to infants that weren’t exposed. Another type of repetitive element, Long Interspersed Nuclear Elements (LINEs), and the mean global methylation has previously been correlated to birth weight percentile. LINE methylation is marker of global (genome wide methylation) and an increase in mean methylation levels for this element, have been directly correlated to an increase in birth weight percentile. These LINE-1 methylation levels are also directly altered due to maternal alcohol use during pregnancy [65] suggesting a potential link between LINE-1 global placental methylation changes, due to alcohol, which results in low-birth-weight infants. Methylation of one gene, *AURKA*, was negatively correlated to fetal weight in the cord blood of full term pregnancies [60]. Differential methylation has been correlated with changes in fetal growth in multiple studies (Table 3) with and without pregnancy co-morbidities. Whilst the correlation with certain clinical data and methylation has been characterised, there remains the question of whether these interactions are a side effect or are contributing to the pathology.

As the primary site of nutrient transfer, perturbations in placental functionality directly impact the developing fetus. Maternal habits and traits have also been linked to adverse fetal outcomes such as an increased incidence of childhood asthma born from asthmatic mothers. There appears to be a pattern of differential methylation occurring in adverse pregnancy outcomes including growth restriction. However, determining the causal factors involved remains difficult. Additionally, identifying whether the methylation changes are contributing to the growth restriction onset or if they are by-products of potential acclimations of the placenta to external modulators is of importance. Previously, the potential of bacterial presence in the placenta has been considered a candidate as an external stimulus due to known interactions between bacteria and host cells. However, this remains under contention due to claims the ‘placental microbiome’ does not exist and the lack of information surrounding the changes that can occur in placental host cells correlating with fetal growth restriction.

## 6. The Controversy Surrounding the Presence of Bacteria in the Placenta

The existence of a true placental microbiome is contentious and recent studies investigating the potential microbiome at this site have suggested that sequenced bacterial DNA is evidence of contamination and does not represent true biological signal. These conclusions are drawn due to the overlap of bacterial DNA with the same sequence homology identified in negative experimental controls and placental samples as well as raw 16S ribosomal RNA (rRNA) counts in biological samples that are similar to negative controls. [79,80,81,82]. However, bacteria have been identified in placental tissue using traditional culture-based techniques [83,84] and, more recently, due to the fastidious requirements of bacterial species that colonise the reproductive tract, identified using molecular-based techniques. These studies have characterised the presence of bacterial DNA in human placental tissue and conclude that bacterial DNA cannot be disregarded as an experimental contaminant alone [85,86,87,88]. Recent re-analysis of two studies that infer that the placenta has no bacterial signature identified the presence of unique bacterial DNA signatures in placental samples that cannot be regarded as contamination as the sequences were found only in placental samples and not in controls [89]. Therefore, despite previous assumptions that the placenta remains sterile throughout pregnancy and any microbiota found within placental tissue is linked to pregnancy complications, it is now clear that this organ harbours evidence of bacterial DNA in the presence and absence of pathology, the origin of which may be seeded from the body’s mucosal sites or may be transient in nature similar to the organ itself [85,88,89,90]. The lack of understanding and characterisation of the placental associated microbial components (such as bacterial DNA) compared to the most well-characterised microbial communities such as those located at mucosal sites (the gut, oral and vaginal mucosa) is attributed to the fact the placenta is a low biomass organ, and microbial DNA at this site may well represent only remnants of exposure rather than living microorganisms. Recent evidence also confirms the presence of microbial communities throughout the continuum of the female genital tract including the fallopian tubes and uterine cavity, previously thought to be sterile like the placenta [91,92,93,94,95,96]. This suggests that the upper genital tract is not sterile and that microbial components may have potential physiological roles in these reproductive niches.

### Microbial DNA Signatures in Healthy and Pathological Placenta

Literature supports a role for bacteria in regulating physiological functions associated with human pregnancy, including increased weight gain and insulin resistance that can be mapped across trimesters [97]. During pregnancy, there are dramatic shifts in the gut microbial community for example, as pregnancy progresses the abundance of specific phyla increase (Proteobacteria and Actinobacteria), overall species diversity decreases and inter-individual variability increases [98]. Changes to the human maternal gut microbiota are also correlated with an increased incidence of preterm birth [99] due to premature rupture of membranes due to intrauterine infection. Evidence also exists for the process of indirect microbial exposure, rather than an actively replicating bacterial community, resulting in host immune priming with microbial DNA through unmethylated bacterial DNA CpG motifs in the human placenta [100]. The placental epigenome may serve as an intermediary between the antigenic stimulus and the inflammatory response leading to adverse pregnancy outcomes [101]. 

Microbiome dysbiosis is emerging as a potential modulator of pregnancy-related pathologies including preeclampsia. The aetiology of preeclampsia is defective placentation; however, infection doubles the risk of developing pre-eclampsia [102]. A small number of studies provide evidence of periopathogenic bacterial DNA (including *Actinobacillus actinomycetemcomitans*, *Fusobacterium nucleatum* sp., *Porphyromonas gingivalis*, *Prevotella intermedia*, *Tannerella forsythensis*, and *Treponema denticola*) in placental tissue collected from preeclamptic women, further supporting a potential role for an infectious aetiology in the multifactorial cause of preeclampsia [103]. Severe chorioamnionitis has also been associated with alterations in the diversity of microbial DNA present in the placenta and placental membranes [104,105]. The existence of bacteria in the preterm placenta, however, remains controversial [79,80,106,107]. In terms of the potential for live bacteria to be contributing to placental pathology in a non-infection, non-inflammatory state, Zheng et al., 2015 [87] reported decreased diversity of the microbial community in the bacterial signatures isolated from placental tissue obtained from low birth weight compared to healthy-weight infants. A study by Antony et al., 2015 [108] investigating the presence of bacterial DNA in 320 human placentas reported differences in gestational weight gain, microbial genera and bacterial metabolism from women who delivered preterm. Perturbations of the oral, vaginal and gut microbiomes have also been associated with preterm delivery, while modulation of the inflammatory response in extremely preterm and SGA infants has been associated with recovery of microorganisms from placental tissue and amniotic fluid using conventional culture techniques [98,109,110,111,112,113,114,115] (Figure 1).

The endogenous maternal microbiota is associated with the development of a functional and adaptive immune system for the fetus. The placenta remains an important immune modulator during pregnancy for the secretion of immunomodulatory proteins that prevent rejection of the fetal allograft. Additionally, a significant proportion of the metabolites identified in the blood are of microbial origin [117] which suggests a possible route of transmission through the transfer of nutrients that occurs between the fetal and maternal circulatory system via the placenta. The presence of microbial DNA and potentially live organisms in placental tissue should be considered as a potential modulator of differential gene expression in the placenta leading to infant pathologies.

## 7. The Potential Microbial–Epigenome–Transcriptome Interactions and Regulation in Placental Health and Pathology

Host–microbe–environment interactions within the placenta form a critical potential stimulus for gene expression changes that may lead to fetal pathologies. These interactions at the maternal-fetal interface likely induce epigenetic and gene expression changes in placental cells resulting in conditions such as fetal growth restriction, preeclampsia and other conditions that are multifactorial. The maternal microbiome is increasingly implicated in human health and disease outcomes, such as fetal growth restriction, via host cell interactions in both mother and baby [98,118,119] supporting a role between the microbiome and infant health. Growth restriction of infants *in utero* has been linked to immune system dysregulation [120] and during pregnancy, the immune system is regulated to balance pregnancy viability via the maternal tolerance of the fetal allograft, with protection from pathogens [121]. Inefficient immunomodulation results in perturbations of placental function and infant pathology (Table 1, Table 2 and Table 3). Transient microbial communities in the placenta may modulate the maternal immune response through microbial DNA or metabolite production affecting the maternal tolerance of the pregnancy and thus nutrient transfer and fetal growth. 

### 7.1. Bacterial Metabolites/DNA and the Potential Influence on Epigenetics in Fetal Growth

Endogenous bacteria play important roles in digestion, immune system function and overall host health via the production of key metabolites including folate, butyrate, lipids and cholesterol that humans cannot synthesise but are required for proper cell function [122,123]. Metabolic programming [124] is where infant gut microbiota changes induce epigenetic modifications that contribute to a body weight phenotype in later life. The direct link between microbiota dysbiosis and an infant phenotype suggests that bacterial metabolites are extremely important for fetal health. Two bacterial metabolites, folate [125] and butyrate [126] are functionally capable of inducing epigenetic changes in surrounding host cells. Microbial components including DNA and metabolic products expressed by transient microbes may be contributing to change in placental epigenetic modification. These components potentially contribute to epigenetic changes within placental tissue and amniotic fluid [127]. Further, postnatal delivery of folate in a mouse model affects DNA methylation of multiple genes in host intestinal cells [128]. DNA methylation in intestinal stem cells is significantly altered in studies of germ-free animals, further supporting the critical role for the host-microbe interaction and intestinal maturation via epigenetic regulation [129]. Butyrate, another product of bacterial metabolism by Gram-positive Firmicutes, affects histone acetylation by inhibiting histone deacetylases in host cells. Histone deacetylases remove acetyl groups on histones thereby inhibiting gene expression through chromatin remodeling. This pathway has been associated with the regulation of multiple genes involved in cholesterol and lipid metabolism and storage [130,131,132]. It is possible that key regulators of metabolism are also epigenetically modified in fetal under-nutrition and over-nutrition due to influence from bacterial metabolites [133,134,135,136].

It is proposed that the underlying modulators of expression are influenced by the bacterial DNA and metabolites present in the placenta. Differential gene expression in the placentae of SGA and LGA infants impacts multiple metabolic and growth pathways (Table 1 and Table 2). Currently, a thorough understanding of the bacterial metabolic activity in the placenta is lacking. However, studies have used bioinformatics programs to predict isolated species with known metabolites. Tryptophan, fatty acid metabolism, and benzoate degradation were predicted metabolic pathways correlating with bacterial species identified by DNA sequencing in the placental tissue in the context of maternal obesity [88]. Tryptophan is a substrate used for the synthesis of the Sirtuin class of histone deacetylases (NAD+ dependent) [137] and placental fatty acid metabolism may include the synthesis of butyrate. 

The link between bacterial components (DNA), metabolites and epigenetic changes indicates the need to more fully elucidate the host-microbe interactions between the endogenous bacterial community and the presence of bacterial DNA and metabolites within the placenta as influencers of fetal pathology. We propose a model of bacterial influence on the epigenetic regulation and expression of genes in fetal pathology (Figure 2). The bacterial components enter the host placental cells and contribute to transcriptional changes via remodeling of the epigenome. Butyrate potentially inhibits histone deacetylases in these cells (enzymes that remove acetyl groups from histone groups, therefore, inhibiting gene expression) resulting in gene over or under-expression. Other bacterial metabolites influence histone deacetylases and epigenetic enzymes including DNA methyltransferases (a group of enzymes that add and remove methyl groups on DNA, therefore, modulate gene expression) resulting in a differential epigenetic profile. These changes then affect the placental genes which are either transcribed or silenced and therefore the corresponding fetal response of key metabolic and growth pathways leading to fetal under and overgrowth dependent on the specific gene involved.

### 7.2. Bacteria Can Modulate Host Cell Expression via Modulation of Epigenetic Marks in Other Models

Site specific microbiomes, including the gut, have recently been correlated with adverse outcomes including colorectal cancer. In a recent review from Allen and Sears 2019 [2], they describe the current body of knowledge linking the presence of specific bacteria to the onset of cancer and other co-morbidities. They describe the potential mechanism of action that is used by microorganisms to modulate the host cell epigenome leading to differential gene expression in a pathology state. These studies provide some of the first evidence that bacteria are capable of and exercise the ability to modulate the human genome in a pathology scenario. For example, these studies identified that common gut commensals *Lactobacillus. acidophilus* and *Bifidobacterium. infantis* can induce DNA hyper and hypo methylation of target genes that may be involved in disease susceptibility [3]. Most of the gut bacteria are cultivable and viable and the relationship between living bacteria and host genome changes are well accepted. As stated previously, however, bacterial metabolites can induce host cell changes and may also been a contender for modulating human health and disease.

This relationship between the microbiota at a host niche and the modulation of the epigenome and transcriptome have been identified in more commonly studied microbiome sites like the gut [129,138,139]. This relationship between cultivatable and viable bacteria has not been characterised in any human reproductive organ or site thus far. Consequently, there is a need for studies to independently validate the change in epigenetics and gene expression in these reproductive sites and then to correlate with microbial sequencing data. Completion of research investigating this multi-omics relationship will further identify the potential contribution of placental derived bacterial metabolites/DNA to the onset of fetal pathology.

## 8. Conclusions

Small-for-gestational-age infants are the result of adverse external modulators, which lead to an increased predisposition to multiple metabolic and cardiovascular conditions via placental dysregulation. The genetic pathways involved in fetal growth restriction are incredibly complex, involve multiple channels of dysregulation and are, currently, not well understood. In addition to gene expression changes regulating metabolism, the immune response and growth pathways, differences in methylation levels of genes and changes in the microbial DNA present in placental tissue have all been correlated with fetal growth restriction. The complex interaction between gene expression, methylation levels, and bacterial DNA, however, requires further elucidation. Future investigation into the modulation of host cell gene expression by bacterial DNA signatures and/or components is therefore required. Important first steps include in vitro studies to determine whether bacterial DNA can modify the epigenome and/or transcriptome in placental cell lines using dominant bacterial signatures from species identified as biologically relevant in placental studies. Trials using DNA isolated from bacteria with variable immunostimulatory properties would support a dose-response analysis and should also be completed. Due to the preliminary evidence presented in the above review, we hypothesise that specific bacterial DNA and/or the corresponding metabolic products may modify DNA methylation and therefore placental gene expression in growth-restricted infants. Characterisation of this relationship could lead to the identification of biomarkers to predict and understand this pathology.

## Figures and Tables

**Figure 1 genes-11-00080-f001:**
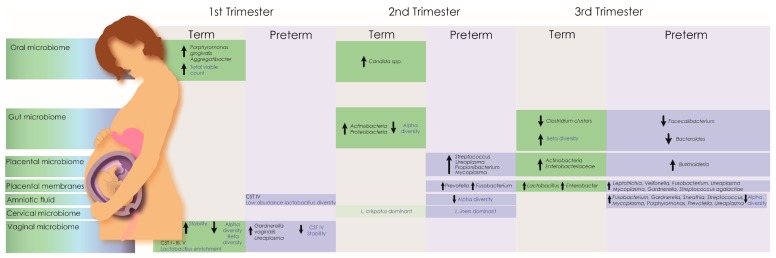
Microbiota associated with preterm and normal birth throughout trimesters. The pregnancy microbiome is in a constant state of flux and no single taxa has proven predictive for adverse pregnancy outcomes. However, the maternal microbiome has been associated with pregnancy risk factors and lifelong infant health outcomes. Changes in the maternal microbiome across multiple anatomical sites are well documented in both term and preterm deliveries. Healthy pregnancy microbiomes generally exhibit increased stability and reduced community diversity, which correlate with increasing gestational age. In infants delivered at term, the vaginal microbiome becomes more stable during the first trimester, dominated by community state types (CST) with highly abundant lactobacilli (CST I (*Lactobacillus crispatus* dominant), II (*Lactobacillus gasseri* dominant), III (*Lactobacillus iners* dominant), V (*Lactobacillus jensenii* dominant) [116]). In contrast, CST IV, characterised by a lack of dominant lactobacilli, is associated with adverse pregnancy outcomes including preterm delivery. During pregnancy, changes in Alpha (species richness and evenness) and beta diversity in the gut and reproductive tract have also been associated with preterm delivery.

**Figure 2 genes-11-00080-f002:**
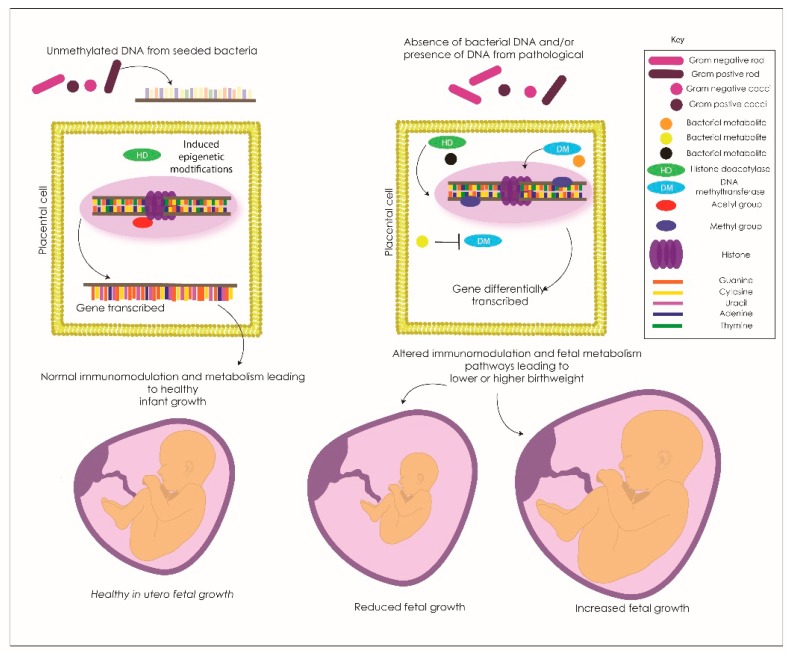
The proposed method of bacterial metabolites and/or bacterial cytosine paired to guanine (CpG) unmethylated DNA interaction with the placental genome that may be modifying placental transcription in growth-restricted infants. The bacterial DNA or metabolites (in this example it is unmethylated bacterial DNA) interact with cell surface receptors on placental cells and the host cell response modifies the epigenome for a specific gene. This interference then allows the specific gene to be transcribed and the normal pathways are expressed contributing to the healthy growth of the placenta and infant in utero.

**Table 1 genes-11-00080-t001:** Gene expression differences in growth-restricted human placenta using RNA-sequencing.

Potential Contributing Factor	Fetal Size	Pathway	Gene Expression Change	Gene	Gene Function	References
	Small for gestational (SGA)	Cell proliferation	Increased	*CREB3*	Transcription factor regulation cell proliferation	[35]
		Protein transport		*DNAJC14*	Regulates export of target proteins from the endoplasmic reticulum to the cell surface	
		Translation/Cellular growth and division		*DDX3X*	Alteration of RNA secondary structure such as translation initiation, nuclear and mitochondrial splicing, and ribosome and spliceosome assembly	
		Development		*GRHL1*	Transcription factor during development	
	SGA	Metabolism	Increased	*QPCT*	An enzyme involved in the synthesis of neuroendocrine peptides	[35]
		Bone formation and haematopoiesis		*FSTL3*	Glycoprotein involved in inhibition of multiple pathways	
		Circadian rhythm		*BHLHE40*	A transcriptional repressor involved in the regulation of the circadian rhythm	
		Gestation, stress response and metabolism		*CRH*	A marker that determines the length of gestation and the timing of parturition and delivery	
		Metabolism	Decreased	*GOT1*	Cytoplasmic form of an enzyme involved in amino acid metabolism and the urea and tricarboxylic acid cycles	
	Large for gestational age (LGA)	Integrity and cell signaling	Decreased	*PLEC*	Member of a family that interlink different elements of the cytoskeleton	
		Cell adhesion		*PVRL4*	Encodes a transmembrane cell adhesion protein comprised of immunoglobulin-like subunits	
		Metabolism and energy		*RDH13*	Mitochondrial enzyme that catalyses the reduction and oxidation of retinoids	
Late-onset preeclampsia	SGA	Multiple roles	Increased	*INHBA*	Involved in both the activation and repression of follicle stimulating hormone as well as a role in eye, tooth and testis development	
		Growth		*PAPPA2*	Regulator of insulin-like growth factor	[37]
		Immune system		*BCL6*	Master regulator which leads the differentiation of naïve helper T cells into follicular helper T cells	
		Cell structure	Decreased	*FAM101B*	Regulates the perinuclear actin network and nuclear shape through interaction with filamins	
		Blood clotting		*F13A1*	A subunit part of the last zymogen to become activated in the blood coagulation cascade	
	Intrauterine growth restriction (IUGR)	Unknown	Increased	*ARMS2*	Unknown function	[37]
		Immune system		*BTNL9*	Involved in class one major histocombability complex mediated antigen pathways	
		Hair morphogenesis		*TCHHL1*	Role in hair morphogenesis and development of complex skin disorders	
		Cell adhesion		*ADAM2*	Roles in cell–cell and cell–matrix adhesions and is an integral complex of sperm cell membranes	
		Epidermal growth factor signaling		*ASTE1*	Nuclease activity and possible role in epidermal growth factor signaling	
		Immune system	Decreased	*THEMIS*	Regulates T-cell selection in late thymocyte development	
		Cellular functions		*PTPRN*	Signaling molecules that regulates cell growth, differentiation, mitotic cycle and oncogenic pathways	
		Inflammation		*FNDC4*	Anti-inflammatory factor in the intestine and acts on macrophages to downregulate pro-inflammatory gene expression	
		Immune system		*SIRPG*	Glycoprotein involved in negative regulation of tyrosine kinase-coupled signaling processes	

**Table 2 genes-11-00080-t002:** Gene expression differences in growth-restricted human placenta using microarrays.

Potential Contributing Factor	Fetal Size	Pathway	Gene Expression Change	Gene	Gene Function	References
	Small for gestational age (SGA)	Metabolism	Increased	*LEP*	Hormone involved in regulating body weight	[41]
		Gestation, stress response and metabolism		*CRH*	A marker that determines the length of gestation and the timing of parturition and delivery	
		Growth factors		*IGFBP-1*	Binds to insulin like growth factors	
Preeclampsia	Fetal growth restriction (FGR)	Angiogenesis	Increased	*FLT1*	Promotes angiogenesis	[42]
				*ENG*	Promotes normal structure and integrity of adult vasculature and regulates the migration of vascular endothelial cell	
	FGR		Decreased	*NRP-1*	Multiple roles in angiogenesis, axon guidance, cell survival, migration, and invasion and implemented in multiple cancers	[43]
	IUGR	Cell growth	Decreased	*PLAGL1q*	Suppresses cell growth	[44]
		Unknown		*GNAS*	Imprinted allele	
		Metabolism		*GATM*	Mitochondrial enzyme that belongs to the amidinotransferase family	
		Growth		*MEG3*	Growth suppressor in tumour cells, Maternally imprinted	
		Enzymatic activity		*MEST*	Loss of imprinting link to cancer growth	
		Growth	Increased	*PHLDA2*	Plays a role in placental growth	
		Gestation, stress response and metabolism		*CRH*	A marker that determines the length of gestation and the timing of parturition and delivery	
		Metabolism		*LEP*	Hormone involved in regulating body weight	
		Inflammatory response		*HPGD*	Metabolism of prostaglandins	
				*SLCO2A1*	Transport of prostaglandins	
	FGR	Immune regulation	Increased	*PSG1*	Roles in allograft tolerance of the fetus	[45]
		Unknown		*PLAC4*	Placental specific expression	
		Neurological		*TAC3*	Excite neurons, evoke behavioural responses, are potent vasodilators and secretagogues, and contract (directly or indirectly) many smooth muscles	
		Unknown		*PLAC3*	Unknown	
		Gestation, stress response and metabolism		*CRH*	A marker that determines the length of gestation and the timing of parturition and delivery	
		Growth		*CSH1*	Involved in stimulating lactation, fetal growth and metabolism	
		Hormones		*KISS1*	Initiates secretion of gonadotropin-releasing hormone as well as other tumour-suppressing roles	
		Metabolism		*CPXM2*	Metalloexopeptidase roles	
	FGR	Cell–cell adhesion	Increased	*CLDN1*	integral membrane protein	[46]
		Protective role	Decreased	*TXNDC5*	induced by hypoxia and its role may be to protect hypoxic cells from apoptosis	
		Cell–cell signaling and protein reuptake and metabolism		*LRP2*	Critical for the reuptake of numerous ligands, including lipoproteins, sterols, vitamin-binding proteins, and hormone	
		Neurological	Increased	*PHLDB2*	involved in the assembly of the postsynaptic apparatus	
	Large for gestational age (LGA)	Cell–cell adhesion	Increased	*CLDN1*	integral membrane protein	
		Metabolism	Decreased	*LEP*	Hormone involved in regulating body weight	
		Neurological		*GCH1*	Involved in making tetrahydrobiopterin a cofactor involved in the production of two neurotransmitters called dopamine and serotonin	

**Table 3 genes-11-00080-t003:** DNA methylation changes that are associated with fetal growth in the human placenta.

Suspected Contributor	Growth Status	Gene/Locus	Hypo/Hypermethylated Compared to Normal Growth Placenta	References
	Intrauterine growth restriction (IUGR) Small for gestational age (SGA)	*ALDH3B2, OAT, CSTA, KLHL5, GPR4, MFAP1, DNAJB4, NCOA4, GYS2, GIMAP2, C20orf28, FGF14, GGPS1, NFKBIZ*	Hypomethylated	[66]
	IUGR/SGA	*APBA2, RPE65, SLC25A18, SERPINA5, MEP1A, PDC, OMG, CHML*	Hypermethylated	
	SGA	*IGF2*	Hypomethylation with decreased expression	[67]
	IUGR	*IGF2*	Hypomethylation with decreased expression	[68]
	SGA	*H19*	Hypomethylation	[67]
	IUGR	*IGF2/H19*	Hypomethylation of the region with a decrease in expression of IGF2	[69]
	IUGR	*SERPINA3*	Hypomethylation with increased expression	[70]
Maternal smoking	Low birth weight	*CYP1A1*	Lower methylation of this locus is correlated with maternal smoking and these maternal smokers had lower-weight infants but no direct correlation between hypomethylation at this locus for low-birth-weight infants	[61]
	Weight gain during the first year	*MEG3* promotor for DLK-1DIO3 imprinted region	Hypomethylation of this promotor is negatively correlated with weight increase during the first year of life	[71]
Preeclampsia	IUGR + preeclampsia	D-loop (mitochondrial DNA)	Lower methylation levels decreased for IUGR/Pre-eclampsia pregnancies	[72]
	IUGR	Metabolic genes—*PCKSKIN, KCNAB2, BLK, PTRN2, GK, GFPTI, AGMO, RPH3AL* Cardiovascular—*PTGIR, NTSRI, ACE, NPY, NOS3, CASZI* Inflammatory—*IL10, CD28, AZUI, IL3RA, MARCH1, LRBA, CD38, IL32*	Differentially methylated between placenta of IUGR and AGA infants	[73]
	Reduced fetal growth	*HSD11B2*	Hypermethylation has been associated with reduced fetal growth	[74]
	Fetal growth restriction (FGR)	*IGF2, AHRR*	Hypomethylation in one position of each locus in FGR placenta	[75]
	FGR	*HSD11B2, WNT2*	Hypermethylation of these locus in FGR	[75]
Maternal asthma	Low-weight infants	*AURKA*	Hypermethylation of this locus correlated with lower fetal height and weight	[60]
	SGA	*RTL1*	Increased methylation in cytosine paired to guanine 1 (CpG1) of the gene	[76]
	Severe SGA	*RTL1*	Very high methylation in CpG1 of the gene	
	SGA	*ATG2B, NKX6.1,SLC13A5*	Hypermethylated in SGA placenta correlated with inverse relationship to gene expression changes	[77]
	SGA	*WNT2*	High promotor methylation	[78]
Early- and late-onset preclamspia	SGA	*FN1, PKM2, KRT15*	Differential methylation	[54]

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
