# Peer review of "Modulation of Placental Gene Expression in Small-for-Gestational-Age Infants"

_genes, 2020, doi:10.3390/genes11010080_

Round 1

Reviewer 1 Report

Review:

“Do placental bacterial metabolites and DNA regulate methylation in growth -restricted infants.”

This title doesn’t reflect the content of the paper or only for a very small amount.

It is a very interesting subject to study as you propose at the end of the paper. But I think you have to rearrange your paper.

It is at page 12 that microbial DNA is started to be discussed. The pages before are very interesting, but are more related to another title : f.i. placental and fetal growth regulation in general. You might also put these (interesting data) in an appendix, it contains as I understand what is known about 57 specific genes and placental and fetal development.

I know how complicated this sort of studies are giving rise to much speculation, since I was involved in studies on effects of toxic chemicals on gene metabolism.

May I propose to publish only pages 12 to 27 and tell about your hypothesis of the importance of the bacterial DNA. And use the other parts for a paper discussing genetic influences in fetal growth in general as an appendix? I love to read all this and very good what you did in the pages 1-12 but that is not important for your message. It can support it.

This might make your paper more readable.

There is a typo: line 215 “characterized”

Reviewer 2 Report

In the present review, O'callaghans and collaborators give actual information about the importance of the microbiota metabolites and the relation with fetal programming specific in the growth fetal. I consider that the original version of the manuscript is clear, objective, substantial, and summarized. However, some changes must be performed. 1) the letter of Figure 1 is small, please modify. 2) The idea in figure 2 is that the bacteria entry in the placenta cell and metabolites bacterial participate in regulating the epigenetic changes; please, modify the bacteria is outside of the cell and could misunderstand the general idea; additionally, the letter is small please change the size of the letter. 3) in figure legend 2, the community state types ( Lines 372-376), please integrate into the paragraph. 4) in conclusion, please, it is essential to give a possible perspective of future research and questions about the research that can provide clear evidence about the subject of this review.

Reviewer 3 Report

I was fine with this review up to the points being made which include the placental microbiome. The presence of a placental microbiome is controversial and there are studies that refute the presence that are not referred to in this manuscript (for example: Kuperman et al, BJOG 2019; Dudley DJ, BJOG 2019; de Goffau et al, Nature 2019;  Theis  et al Am J Obstet Gynecol 2019, and others).  The authors should discuss the controversy and be forthright about their opinion that there is a placental microbiome despite these published results.

Round 2

Reviewer 1 Report

Dear author

I agree now completely with the revised version of the paper 

Reviewer 3 Report

Thank you for a thoughtful reply to my concerns.